# Spatial integration of sensory input and motor output in *Pseudomonas aeruginosa* chemotaxis through colocalized distribution

**Zhengyu Wu[1,2,3†], Maojin Tian[4†], Sanyuan Fu[2†], Min Chen[2], Rongjing Zhang[2]\*, Junhua Yuan[2]\***

[1]Research Center of Translational Medicine, Central Hospital Affiliated to Shandong First Medical University, Jinan, China; [2]Hefei National Research Center for Physical Sciences at the Microscale and Department of Physics, University of Science and Technology of China, Hefei, China; [3]Research Center of Translational Medicine, Jinan Central Hospital, Shandong University, Jinan, China; [4]Department of Critical Care Medicine, Zibo Central Hospital Affiliated to Binzhou Medical University, Zibo, China

**\*For correspondence:**
rjzhang@ustc.edu.cn (RZ);
jhyuan@ustc.edu.cn (JY)

[†]These authors contributed equally to this work

**Competing interest:** The authors declare that no competing interests exist.

## eLife Assessment

This **important** study by Wu et al presents **convincing** data on bacterial cell organization, demonstrating that the two structures that account for bacterial motility - the chemotaxis complex and the flagella - colocalize to the same pole in *Pseudomonas aeruginosa* cells, and expose the regulation underlying their spatial organization and functioning. This manuscript will be of interest to cell biologists, primarily those studying bacteria.

**Abstract** The opportunistic pathogen *Pseudomonas aeruginosa* serves as a model organism for studying multiple signal transduction pathways. The chemoreceptor cluster, a core component of the chemotaxis pathway, is assembled from hundreds of proteins. The unipolar distribution of receptor clusters has long been recognized, yet the precise mechanism governing their assembly remains elusive. Here, we directly observed the relative positions of the flagellar motor and chemoreceptor cluster using flagellar filament labeling and gene editing techniques. Surprisingly, we found that both are located at the same cell pole, with the distribution pattern controlled by the polar anchor protein FlhF. Additionally, the efficient assembly of the chemoreceptor cluster is partially dependent on the integrity of the motor structure. Furthermore, we discovered that overexpression of the chemotaxis regulatory protein CheY leads to high intracellular levels of the second messenger c-di-GMP, triggering cell aggregation. Therefore, the colocalization of the chemoreceptor cluster and flagellum in *P. aeruginosa* serves to avoid cross-pathway signaling interference, enabling cells to conduct various physiological activities in an orderly manner.

## Introduction

In biological systems, the process by which proteins self-assemble into organized complex structures is widespread (*Halatek et al., 2018*; *Thanbichler and Shapiro, 2008*). Pole-to-pole protein oscillations in the *Min* system ensure that the FtsZ ring, a crucial component of cell division, is placed precisely in the middle of the cell body (*Loose et al., 2011*; *Rothfield et al., 2005*). In response to antibiotic

treatment and heat stress, some bacteria generate multiple protein foci throughout the cytoplasm (*Fay and Glickman, 2014*; *Vaubourgeix et al., 2015*). The polar localization of the secretion system, such as the type VI secretion system, is mediated by targeting proteins and potentially facilitates host-pathogen interactions (*Wang et al., 2019*; *Low et al., 2014*). Thus, spontaneous assembly is accompanied by the formation of spatial patterns that organize the intracellular environment.

To grow and colonize in various ecological niches, bacteria have evolved a mechanism for migrating towards more favorable environments. The locomotion organ (the flagellar motor) and the chemotaxis pathway play crucial roles in achieving this goal. The former provides physical drive, while the latter offers directional guidance. The initiation of cell reorientation is controlled by motor switching, which is governed by the chemotaxis two-component signaling pathway. This pathway comprises transmembrane chemoreceptors, a histidine kinase CheA, an adaptor protein CheW, a response regulator CheY, and two adaptation proteins, CheB and CheR. Receptors collaborate with CheA and CheW to form a chemotaxis complex in a hexagonal pattern (*Briegel et al., 2012*; *Zhang et al., 2007*). Upon phosphorylation of CheY by the kinase at the complex, it freely diffuses to the motor and binds to FliM, a component of the motor switch complex, to affect motor switching (*Cluzel et al., 2000*; *Sourjik and Berg, 2002*; *Welch et al., 1993*). It is worth noting that there exists diversity among different bacterial species in the placement of flagellar motors (*Schuhmacher et al., 2015*), whereas chemoreceptor clusters are typically found at one or both ends of the cell (*Jones and Armitage, 2015*). These two structural units are functionally connected. However, it remains unclear whether there is an interaction in their spatial distribution and what the specific regulatory mechanism might be.

*Pseudomonas aeruginosa*, a common human opportunistic pathogen, possesses four chemosensory pathways that perform distinct functions and are stimulated by signal binding to 26 chemoreceptors (*Matilla et al., 2021*). Among them, the proteins encoded by chemotaxis-related genes collectively constitute the F6 pathway. Previous studies have suggested that the receptors involved in this pathway localize to the old cell pole (*Güvener et al., 2006*), similar to the flagellum (*Amako and Umeda, 1982*), implying colocalization of the receptors and the flagellum at the same pole; however, direct evidence for this colocalization is still lacking. Furthermore, the polar anchor protein FlhF has been reported to guide the flagellum to grow at the cell pole (*Schniederberend et al., 2013*; *Murray and Kazmierczak, 2006*). However, the mechanism by which chemotaxis complex-related proteins aggregate and self-assemble at the cell pole is controversial. The distribution of the chemotaxis complex in the peritrichous bacterium *Escherichia coli* has been extensively studied and attributed to several mechanisms, including stochastic self-assembly (*Greenfield et al., 2009*; *Thiem and Sourjik, 2008*), membrane curvature sorting (*Strahl et al., 2015*; *Draper and Liphardt, 2017*), and inefficient clustering in the lateral region (*Koler et al., 2018*). However, these mechanisms cannot explain the unipolar distribution pattern observed in *P. aeruginosa*. Unlike *E. coli*, specific genes responsible for the placement of the chemotaxis complex have been identified in several bacterial species. In *Caulobacter crescentus*, chemotaxis complex assembles at the new cell pole with the help of TipN and TipF proteins (*Huitema et al., 2006*). A tripartite ParC-ParP-CheA interaction network was reported to promote polar localization of chemotaxis complex in *Vibrio parahaemolyticus* (*Ringgaard et al., 2011*; *Ringgaard et al., 2014*). However, as there are no related genes in *P. aeruginosa*, the mechanism of its chemotaxis complex distribution remains a mystery.

Here, we combined gene editing and in vivo fluorescence imaging of flagellar filaments to directly observe the distribution of chemotaxis complex and flagellar motors, proposing a cooperative construction model of chemotaxis network and flagella during the entire division cycle of *P. aeruginosa*. The core focus of this study is to clarify the regulatory mechanism of its chemotaxis complex distribution. We found a substantial association between the assembly of the flagellar motor and the chemotaxis complex. The assembly efficiency of the receptor complex is influenced by core flagellar motor components, particularly the C-ring and MS-ring structures, while its assembly site is also affected by the polar anchor protein FlhF. Furthermore, by introducing exogenously expressed CheY protein, we found that this triggers the expression of c-di-GMP at a high level. From this, we infer that the colocalization of the chemotaxis complex and the flagellum in *P. aeruginosa* avoids the cross-pathway interference of signaling molecules, thus providing a guarantee for the coexistence of multiple chemosensory pathways.

## Results

### Robust generation of daughter cell with both chemotaxis network and flagellar motor

CheY has been shown to colocalize with chemoreceptors (*Güvener et al., 2006*; *Sourjik and Berg, 2000*). To visualize the distribution of chemotaxis complex in cells, we fused the gene encoding the enhanced yellow fluorescent protein (eYFP) to the *cheY* gene in the *P. aeruginosa* chromosome (*Figure 1A*). All genetic modifications were carried out at the native position on the chromosome, allowing the expression of chemotaxis proteins in precise stoichiometry from their natural promoters (*Figure 1B*). The mutant strain with *cheY-eyfp* was able to moderately expand on soft agar plates (*Güvener et al., 2006*), proving that its chemotaxis behavior was not notably affected, so it was selected for subsequent experiments. As shown in *Figure 1C*, CheY-eYFP was mainly located at the cell pole, suggesting that CheA, CheW, and CheY in *P. aeruginosa* form a signal transduction complex that was primarily distributed at the cell pole, as described previously (*Güvener et al., 2006*). Representative large-field images containing many cells are shown in *Figure 1—figure supplement 1*. Additionally, we constructed a plasmid expressing CheA-CFP and electroporated it into the *cheY-eyfp* strain. Fluorescence imaging revealed a high degree of spatial overlap between CheA-CFP and CheY-EYFP (*Figure 1—figure supplement 2*), confirming that CheY-EYFP accurately marks the location of the chemoreceptor complex. From our measurements, nearly 90% (332/372) of cells contain obvious receptor clusters, and factors such as fluorescence bleaching may have caused the loss of fluorescent spots in some cells. In addition, we observed the presence of fluorescent spots at both ends of some cells with a large aspect ratio (probably approaching cell division), indicating that the newly generated progeny cells will have a complete chemotaxis network. Next, we sought to observe the distribution of the flagella in cells simultaneously to understand the motility of future progeny.

Our recent study demonstrated the successful labeling of the flagellar filament with thiol-reactive fluorescent dye by introducing cysteines into the flagellin FliC (*Wu et al., 2021*; *Tian et al., 2022*; *Figure 1B*). To avoid fluorescence interference, we utilized a dye with an excitation peak near 568 nm, considering that the excitation peak of eYFP is 513 nm and the emission peak is 527 nm. We employed a xenon lamp as the excitation light source and switched filters to enable simultaneous fluorescence imaging of flagellar filaments and chemotaxis complex within the same cell.

Remarkably, our findings revealed a surprising colocalization of chemotaxis complex and flagellar filaments at the same end of the cell body, and this colocalization was consistent, as shown in *Figure 1D*. A typical large field was shown in *Figure 1—figure supplement 3*. Similar to the chemotaxis complex, we observed the phenomenon of double flagella symbiosis in cells with a large aspect ratio, suggesting that *P. aeruginosa* has evolved a cell division mechanism with precise timing regulation. Before cell division, both poles of the mother cell assembled chemotaxis complexes and flagellar motors. This ensures that daughter cells possess complete motility and chemotaxis, thereby greatly enhancing the environmental adaptability of the population.

### FlhF controls polar targeting but not flagellum-chemoreceptor colocalization

Despite potential differences in the physical and especially physiological environments at the two cell poles, it is unlikely that the unipolar distribution of the chemotaxis complex can be attributed to passive regulatory factors. The number and distribution of flagellar motors and chemotaxis complexes vary among different bacterial species (*Schuhmacher et al., 2015*; *Jones and Armitage, 2015*), and the localization system for flagella has been well studied. Specifically, the FlhF-FlhG system, discovered in several monotrichous bacterial species, has been shown to control the location and number of flagella (*Altegoer et al., 2014*; *Kazmierczak and Hendrixson, 2013*). In *P. aeruginosa*, a knockout of *flhF* leads to mis-localized flagellar assembly (*Schniederberend et al., 2013*; *Murray and Kazmierczak, 2006*). Considering the consistent colocalization pattern between chemotaxis complex and flagellar motors in *P. aeruginosa*, we speculate that the distribution of chemotaxis complex is also regulated by similar molecular mechanisms.

To investigate the role of FlhF in the localization of chemotaxis complex, we constructed a Δ*flhF* strain for fluorescence observation. The results revealed that the chemotaxis complex no longer grow robustly at the cell pole (*Figure 2A*), and the assembly positions of the flagellar motor change

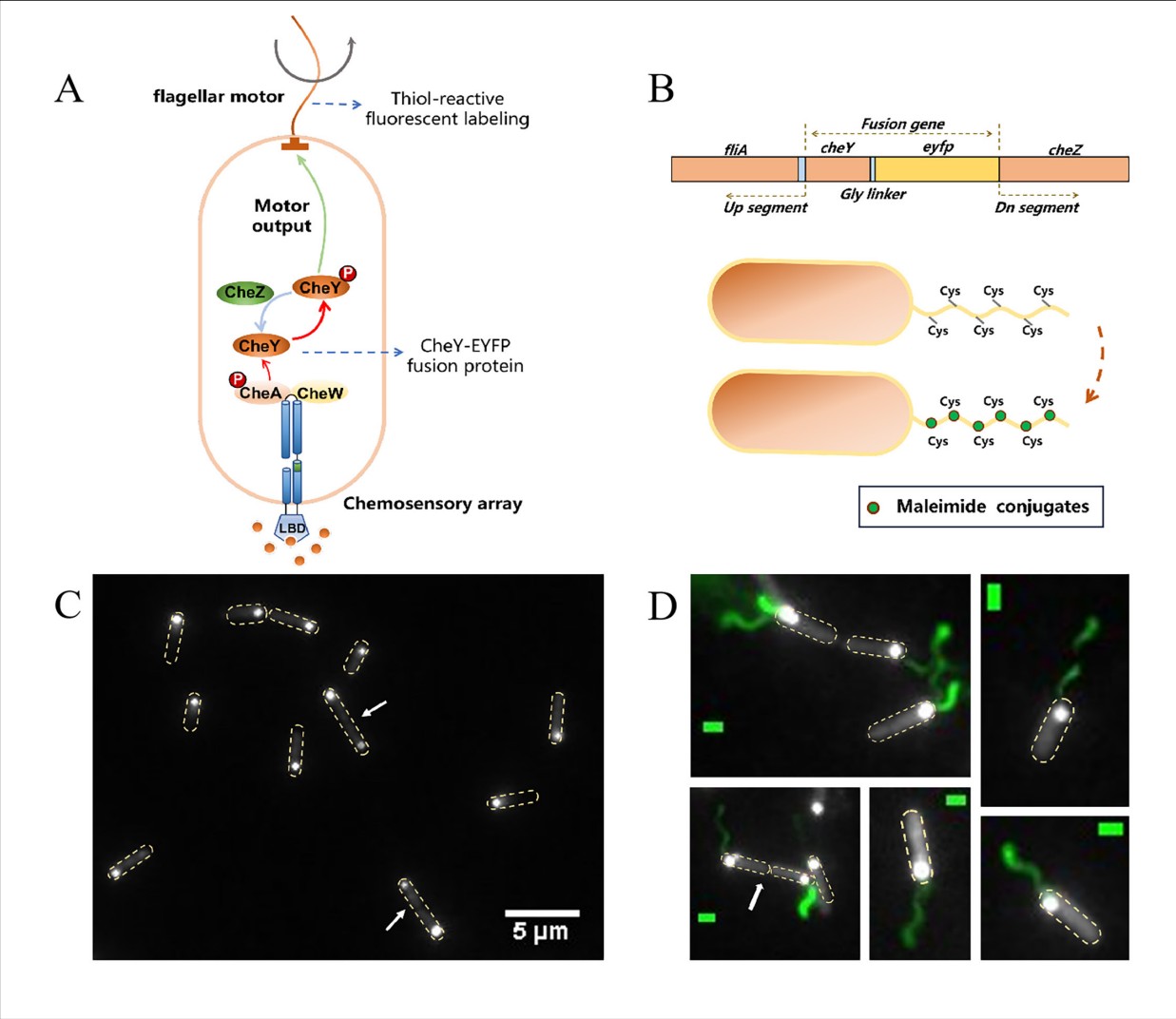

**Figure 1.** Spatial distribution of the chemotaxis complex and flagellar motor in wild-type *P. aeruginosa*. (**A**) Schematic diagram of the chemotaxis signal transduction network and flagellar motor of *P. aeruginosa*. Fusion protein CheY-EYFP and fluorescently labeled flagellar filaments were used as markers to indicate the position of the chemotaxis complex and motor, respectively. (**B**) The labeling mechanism of flagellar filaments and chemotaxis regulatory protein CheY. Filaments (with cysteine point mutation FliC$^{T394C}$) were labeled through sulfhydryl-maleimide conjugation, and *cheY-eyfp* fusion with a 3x glycine linker was used to visualize chemotaxis complex positions. (**C**) Localization of CheY-EYFP in the wild-type strain of *P. aeruginosa*. CheY-EYFP is mainly located at the single cell pole, and the white arrow points to individuals with an obvious chemotaxis complex at both cell poles, which generally have a large aspect ratio of the cell body. The yellow dashed box marks the cell outline. (**D**) The merged imaging of flagellar filaments and CheY-EYFP in the wild-type strain of *P. aeruginosa*, where flagellar motor and chemotaxis complex colocalize in cells. 145 cells with labeled flagella were observed, all of which exhibited consistent colocalization. White arrows point to individuals about to be divided, and the yellow dashed box marks the cell outline. The scale bar is 1 μm.

The online version of this article includes the following figure supplement(s) for figure 1:

**Figure supplement 1.** Distribution of chemotaxis complexes in multiple strains within representative large fields.

**Figure supplement 2.** Co-localization of CheY-EYFP and CheA-ECFP in *P. aeruginosa*.

**Figure supplement 3.** Simultaneous observation of the chemotaxis complex and flagellar filaments in the wild-type strain, shown in a representative large field.

accordingly, colocalizing with the chemotaxis complex (*Figure 2B*). Representative large fields containing many cells are shown in *Figure 1—figure supplement 1*. Additionally, we categorized receptor cluster distribution patterns into three types: precise polar, near polar, and mid-cell. We artificially divided the intracellular area along the long axis of the cell. The two ends of the length accounted for 10% each, which we called the precise-polar domains, the middle 50% became the

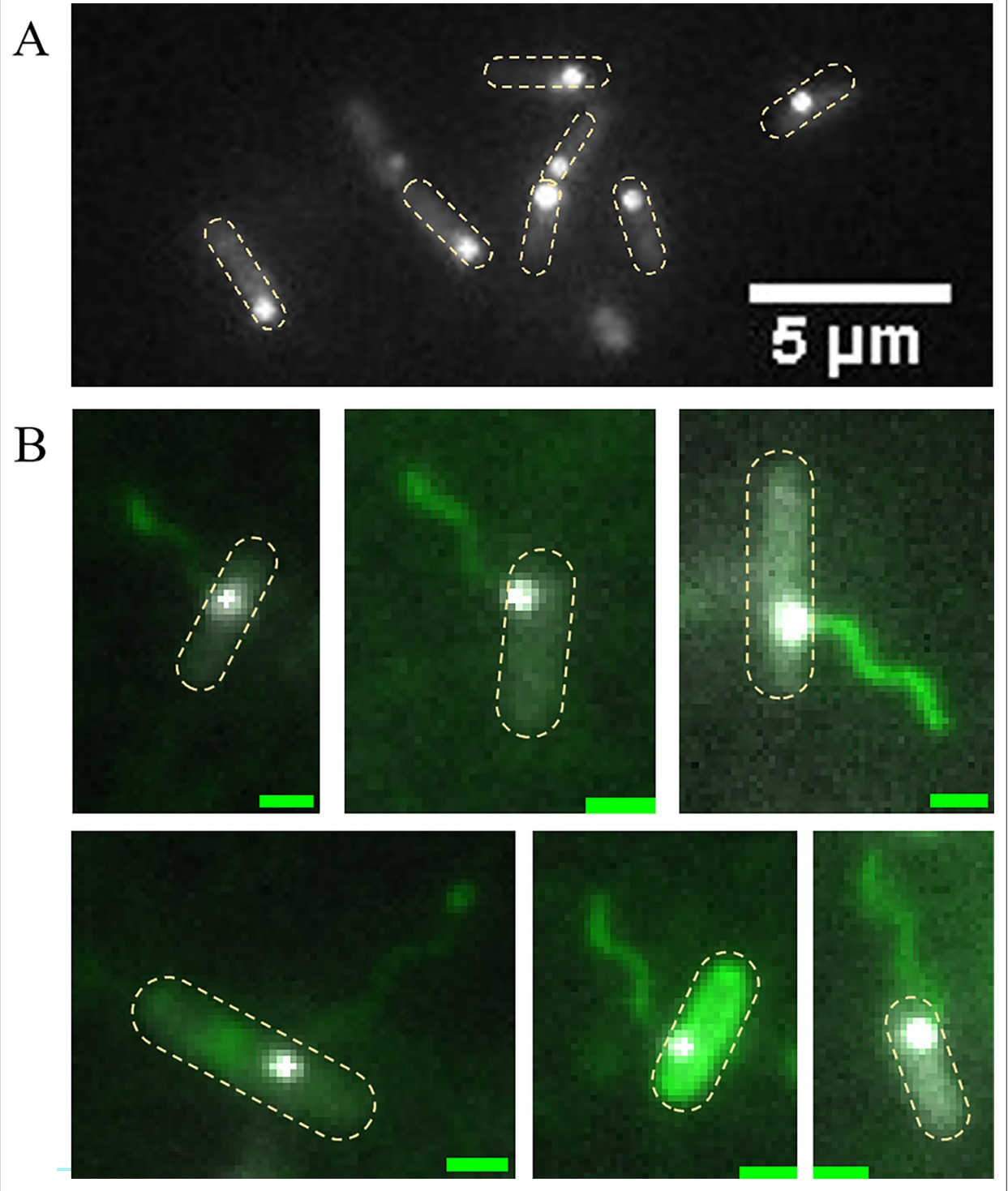

**Figure 2.** Spatial distribution of the chemotaxis complex and flagellar motor in ΔflhF mutant. (**A**) Localization of CheY-EYFP in the ΔflhF strain of *P. aeruginosa*. CheY-EYFP is no longer robustly distributed at the single-cell pole. The yellow dashed box marks the cell outline. (**B**) The merged imaging of flagellar filaments and CheY-EYFP in the ΔflhF strain of *P. aeruginosa*, flagellar motor and chemotaxis complex still colocalize in cells. 101 cells with labeled flagella were observed, all of which exhibited consistent colocalization. The yellow dashed box marks the cell outline. The scale bar is 1 μm.

The online version of this article includes the following figure supplement(s) for figure 2:

**Figure supplement 1.** Quantitative statistics of chemotactic complex distribution in wild-type and flhF-related mutants.

mid-cell domain, and the rest was called the near-polar domain. The type of complex distribution was determined by the partition where the center of the chemotactic complex fluorescent bright spot was located. The relevant statistics for the wild-type strain and the Δ*flhF* mutant are presented in *Figure 2—figure supplement 1*, showing that the proportion of precise polar distribution of chemotactic complexes decreased by 12.8% after *flhF* knockout. We also quantified the proportion of individuals with obvious receptor clusters, which was more than 80% (182/221), similar to the wild-type strain. These experimental results suggest that the polar anchoring protein FlhF has a minimal impact on the assembly efficiency of *P. aeruginosa*'s chemotaxis complex, but it affects its unipolar distribution. Furthermore, a consistent colocalization between the flagellar motor and chemotaxis complex was observed, independent of FlhF. Thus, it is crucial to determine whether there is a causal relationship in the assembly order of these two structural units.

## Core motor structures influence chemoreceptor complex assembly

Cryo-electron microscopy has successfully revealed the complete flagella structure in various bacterial species. This structure includes multiple components such as the inner-membrane MS ring, the cytoplasmic C ring, and the internal protein secretion system, all of which exhibit strong similarities (*Minamino and Imada, 2015*). This suggests that the core structure of the flagellar motor is highly conserved. The MS ring, composed of 26 FliF proteins, forms the foundation of the flagellar structure (*Suzuki et al., 2004*). In addition to serving as a mounting platform for the C ring, the MS ring also acts as a protective shell for the internal protein secretion machinery (*Minamino et al., 2008*). FliG, the main component protein of the C ring, binds directly to the cytoplasmic surface of FliF in a 1:1 ratio (*Levenson et al., 2012*). Thus, the efficient assembly of the C ring requires preassembly of the MS ring. To observe the distribution of the chemotaxis complex following the disruption of the C ring and MS ring, we constructed Δ*fliG* and Δ*fliF* mutants.

We performed fluorescence observation on the Δ*fliG* mutant, which has a disrupted flagellar motor C ring. The results showed that the proportion of Δ*fliG* cells with obvious chemoreceptor clusters decreased significantly compared to the wild-type cells (59.8%, 140/234), although the chemotaxis complex remained at a single cell pole. We also examined the Δ*fliF* mutant, which has a disrupted flagellar motor MS ring. Similar to the Δ*fliG* mutant, the proportion of individuals with obvious chemotaxis complex decreased significantly (62.5%, 202/323). Subsequently, we constructed a Δ*flhF*Δ*fliF* mutant and observed that the proportion of individuals with obvious chemoreceptor clusters further decreased (50.7%, 341/672). To ascertain whether it is motor integrity rather than functionality that influences the efficiency of chemotaxis complex assembly, we constructed a stator mutant (Δ*motA*Δ*motCD*). In this mutant, the motor is completely stalled while the structure remains intact. We found that the mutant performed similarly to the wild-type strain in terms of chemotaxis complex assembly (84.3% of individuals with obvious clusters, 204/242). The fluorescence imaging of receptor clusters for multiple mutants in this section is shown in *Figure 3A*, and corresponding large-field images are provided in *Figure 1—figure supplement 1*. We utilized Western blotting to measure the expression levels of CheY, which were found to be similar across the different strains (*Figure 3B*). These further substantiated that the observed phenomenon is based on the structural integrity of the motor rather than the protein expression level. In contrast, the *cheA* knockout strain was constructed to disrupt the assembly of the chemoreceptor complex. We fluorescently labeled its flagellar filaments and found that its phenotype was no different from that of the wild-type strain (*Figure 3—figure supplement 1*), indicating that the assembly of the chemotactic receptor complex does not affect the flagellar assembly efficiency. The P ring (mainly composed of FlgI) is thought to act as a bushing for the peptidoglycan layer, and its absence results in partial assembly of the motor structure (*Hizukuri et al., 2006*). We constructed Δ*flgI* mutant and found that the proportion of cells with distinct chemotactic complexes was similar to that of the wild-type (*Figure 3—figure supplement 2*). Additionally, we quantified the proportion of cells with receptor clusters (*Figure 3C*). Overall, our findings suggest that the polar anchor protein FlhF and the structural integrity of the motor are crucial for the formation of chemotaxis complex. The former guides them to the appropriate site, while the latter influences the assembly efficiency of the chemoreceptor complex.

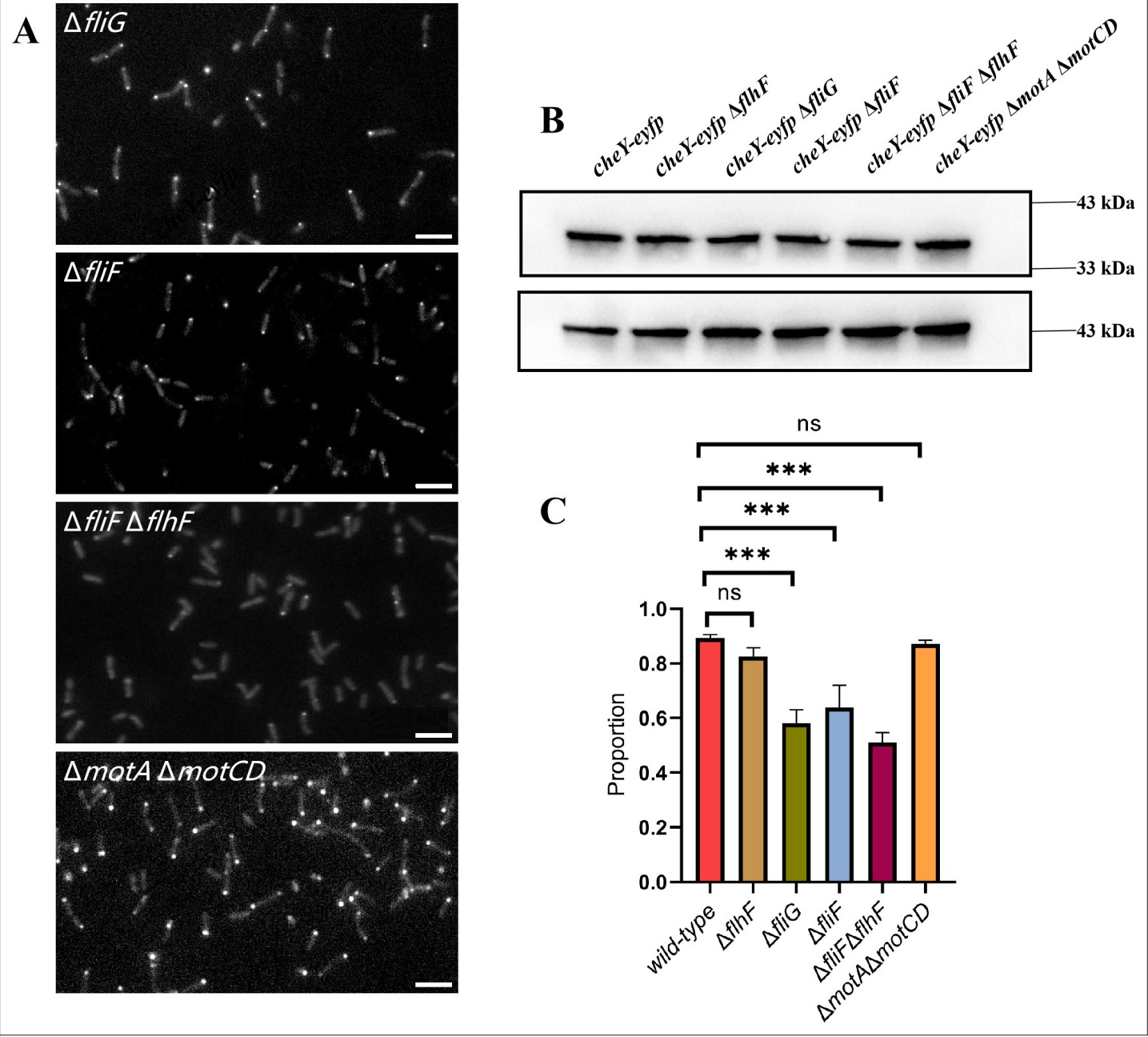

**Figure 3.** Characterization of chemotactic complex distribution in several mutants with incomplete flagellar motor. (**A**) Localization of CheY-EYFP in various *P. aeruginosa* strains. The scale bar is 10 μm. (**B**) Western blot analysis was performed to detect CheY expression in various *P. aeruginosa* strains. β-actin was used as the housekeeping protein. (**C**) Occurrence probability of obvious chemotaxis complex in *P. aeruginosa* wild-type and several mutant strains. The proportion of individuals with obvious chemotaxis complex decreased significantly in the motor-incomplete strains (Δ*fliF* and Δ*fliG*), and this value was further reduced after *flhF* knockout (Δ*fliF*Δ*flhF*). The Δ*flhF* and Δ*motA*Δ*motCD* strains have a similar chemotaxis complex occurrence probability as the wild-type strain. The number of cells analyzed for each strain (from left to right) were: 372, 221, 234, 323, 672, and 242. '\*\*\*': significant difference (p-value<0.0001), 'ns.': no significant difference (p-value>0.05).

The online version of this article includes the following source data and figure supplement(s) for figure 3:

**Source data 1.** PDF file containing original western blots for *Figure 3B*, indicating the relevant bands and treatments.

**Source data 2.** Original files for western blot analysis displayed in *Figure 3B*.

**Figure supplement 1.** Knockout of *cheA* did not affect flagellar assembly efficiency of *P. aeruginosa*.

**Figure supplement 2.** Localization of CheY-EYFP in the Δ*flgI* strain of *P. aeruginosa*.

## Colocalization of chemotaxis complex and flagellar motor avoids cross-pathway interference in *P. aeruginosa* signal transduction

The distribution patterns of receptors associated with various signal transduction pathways in *P. aeruginosa* vary considerably. For instance, the biofilm formation-related receptor WspA, unlike the chemotaxis receptors discussed here, is distributed throughout the cell. This distribution is thought to enhance its sensitivity to mechanical perturbations of the cell membrane (*O'Connor et al., 2012*). Previous studies speculated that the highly consistent position of chemotaxis complex and flagellar motors enhances bacterial chemotaxis performance (*Ringgaard et al., 2014*). However, the diffusion time of the phosphorylated chemotaxis regulatory protein CheY-P across the longest distance in a bacterial cell body (along the cell's long axis) is approximately 100 ms (*Terasawa et al., 2011*; *Sagawa et al., 2014*), whereas the timescale for the chemotaxis temporal comparison is on the order of seconds (*Segall et al., 1986*). Additionally, a study by Fukuoka and colleagues reported that intracellular chemotaxis signal transduction requires approximately 240 ms beyond CheY or CheY-P diffusion time (*Sagawa et al., 2014*). Moreover, the CCW/CW interval of the *P. aeruginosa* flagellar motor under normal conditions is 1–2 s, as determined by bead assays (*Wu et al., 2021*) or tethered cell assays (*Qian et al., 2013*). Taken together, these indicate that for *P. aeruginosa*, which moves via a run-reverse mode, the potential 100 ms reduction in response time due to co-localization of the chemotaxis complex and motor has a limited effect on overall chemotaxis timing. Consequently, the physiological significance of the colocalization of chemoreceptors and flagellar motors remains unclear.

*E. coli* mediates chemotaxis through a single pathway involving five types of chemoreceptors (*Sourjik, 2004*). As a core regulatory protein for chemotaxis, CheY participates in multiple processes such as adaptation and chemotaxis response, with CheY-P binds to the motor to regulate switching (*Cluzel et al., 2000*). In contrast, *P. aeruginosa* has four distinct chemosensory pathways that perform different functions, involving different CheY homologs, and are stimulated by signals binding to 26 types of chemoreceptors. Consequently, *P. aeruginosa* possesses more complex chemosensory pathways (*Matilla et al., 2021*). We thus hypothesized that the co-localization of chemoreceptors and flagellar motors in *P. aeruginosa* ensures locally distributed CheY-P molecules, thereby eliminating the need for a high level of intracellular CheY-P and avoiding potential side effects on other signaling pathways.

To test the potential effect of an increased intracellular CheY-P level, we constructed the CheY expression plasmid *cheY*-pJN105, transformed it into the wild-type *P. aeruginosa* strain, and induced it with varying concentrations of arabinose. We observed that the higher the inducer concentration, the more pronounced the cell aggregation became in the field of view (*Figure 4A*). The transition from planktonic individuals to aggregated communities is similar to biofilm formation, which is known to be accompanied by a significant increase in intracellular c-di-GMP levels.

To investigate whether a higher level of intracellular CheY-P increased the intracellular c-di-GMP level, we sought to detect the c-di-GMP level. We introduced the plasmid pCdrA-gfp into *P. aeruginosa*, using it as a c-di-GMP biosensor with the fluorescence intensity proportional to the c-di-GMP level (*Rybtke et al., 2012*). To ensure the coexistence of the plasmids, the intracellular CheY concentration was controlled by the plasmid *cheY*-pME6032. The intracellular CheY levels induced by varying arabinose concentrations were measured, showing clear differences across the concentration gradient (*Figure 4—figure supplement 1*). We found that the average single-cell fluorescence intensity of the CheY overexpression strain was 71.8% higher than that of the wild-type strain, while the c-di-GMP level of the *cheY* deletion strain was 58.6% lower than that of the wild-type (*Figure 4B*).

The colocalization of both the signaling source (the kinase) and sink (the phosphatase) at the chemoreceptor complex at the cell pole results in a rapid decay of CheY-P concentration within approximately 0.2 μm from the cell pole, leading to a nearly uniform distribution elsewhere in the cell, as demonstrated by *Vaknin and Berg, 2004*. This spatial arrangement effectively confines high CheY-P levels to the pole region. When the motor is also localized at the cell pole, the need for elevated CheY-P concentrations throughout the cytoplasm is reduced. Therefore, the colocalization of chemotaxis complex and flagellar motor facilitates the precise regulation of cell swimming direction within the low intracellular CheY-P concentration threshold, using locally distributed CheY-P molecules. This helps to avoid the occurrence of the aforementioned cross-pathway interference. To further confirm that CheY overexpression promotes aggregation through increased c-di-GMP levels, we performed

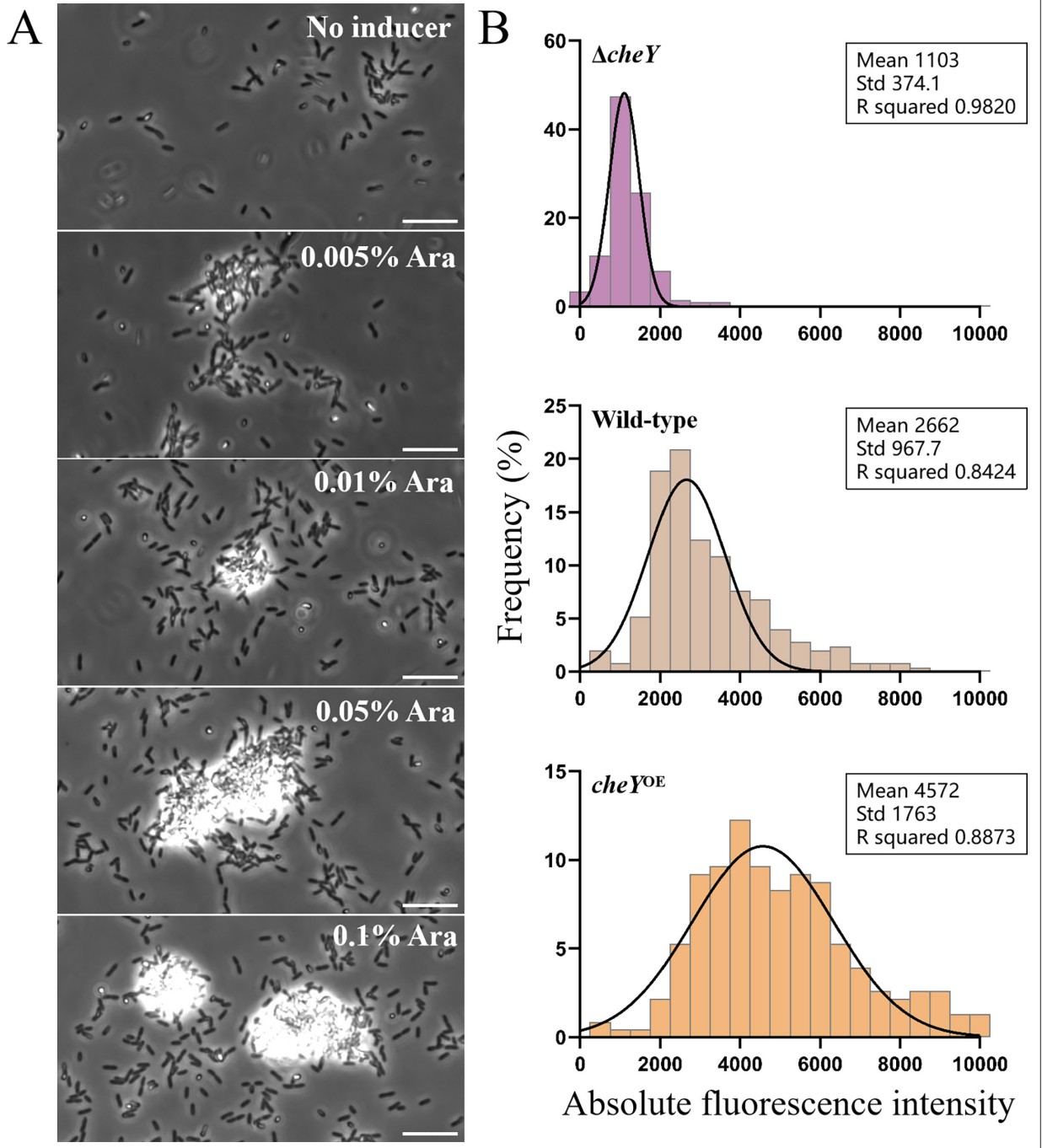

**Figure 4.** Increased CheY content elevates intracellular c-di-GMP levels, leading to cell aggregation. (**A**) The evolution of cell aggregation as the intracellular CheY concentration increases by induction with higher concentrations of arabinose. The scale bar is 10 μm. (**B**) Quantitative characterization of intracellular c-di-GMP levels at different CheY concentrations. From top to bottom, they correspond to the Δ*cheY* strain (N=198), wild-type strain (N=249), and CheY overexpression strain (N=228), respectively.

The online version of this article includes the following source data and figure supplement(s) for figure 4:

**Figure supplement 1.** Intracellular CheY levels induced by different arabinose concentrations, measured by western blot analysis, showing clear differences across the concentration gradient.

**Figure supplement 1—source data 1.** PDF file containing original western blots for *Figure 4—figure supplement 1*, indicating the relevant bands and treatments.

**Figure supplement 1—source data 2.** Original files for western blot analysis displayed in *Figure 4—figure supplement 1*.

*Figure 4 continued on next page*

*Figure 4 continued*

**Figure supplement 2.** Co-overexpressing CheY and the phosphodiesterase (PDE) YhjH from *E. coli* mitigates cell aggregation caused by CheY overexpression in *P. aeruginosa*.

additional experiments co-overexpressing CheY and a phosphodiesterase (PDE) from *E. coli* to reduce intracellular c-di-GMP. These experiments showed that PDE expression mitigates cell aggregation caused by CheY overexpression in *P. aeruginosa* (*Figure 4—figure supplement 2*).

## Discussion

*P. aeruginosa* harbors multiple signal transduction systems that regulate flagella-mediated swimming motility (Che pathway), pili-mediated interface twitching motility (Pil pathway), and biofilm formation (Wsp pathway) (*Matilla et al., 2021*). These systems allow it to thrive in various ecological niches within complex external environments. Here, employing a combination of chromosomal fluorescent protein fusion and flagellar labeling techniques in living cells, we directly observed that the chemotaxis complex of *P. aeruginosa* Che pathway and the flagellar motor share a high degree of consistency in their assembly sites. We found that the chemotaxis complex, comprising Che proteins, was consistently located at the cell pole where the flagellum was positioned. Based on these observations, we deduced the construction mode of the chemotactic network and flagellar motor during the entire cell growth cycle (*Figure 5A*). As the cell body matures and elongates, a new flagellum grows at the opposing cell pole, accompanied by the assembly of a fresh chemotaxis complex. The cell then undergoes division from the middle, generating two daughter cells, each equipped with fully functional motility and chemotaxis capabilities. This ensures robust inheritance of these chemotaxis and motility-related macromolecular machines.

Based on our understanding of the regulation mechanism of flagellar position in *P. aeruginosa*, we constructed a Δ*flhF* mutant and found that the position distribution of its chemotaxis complex is actively regulated by FlhF-mediated molecular mechanisms, rather than being passively restricted by

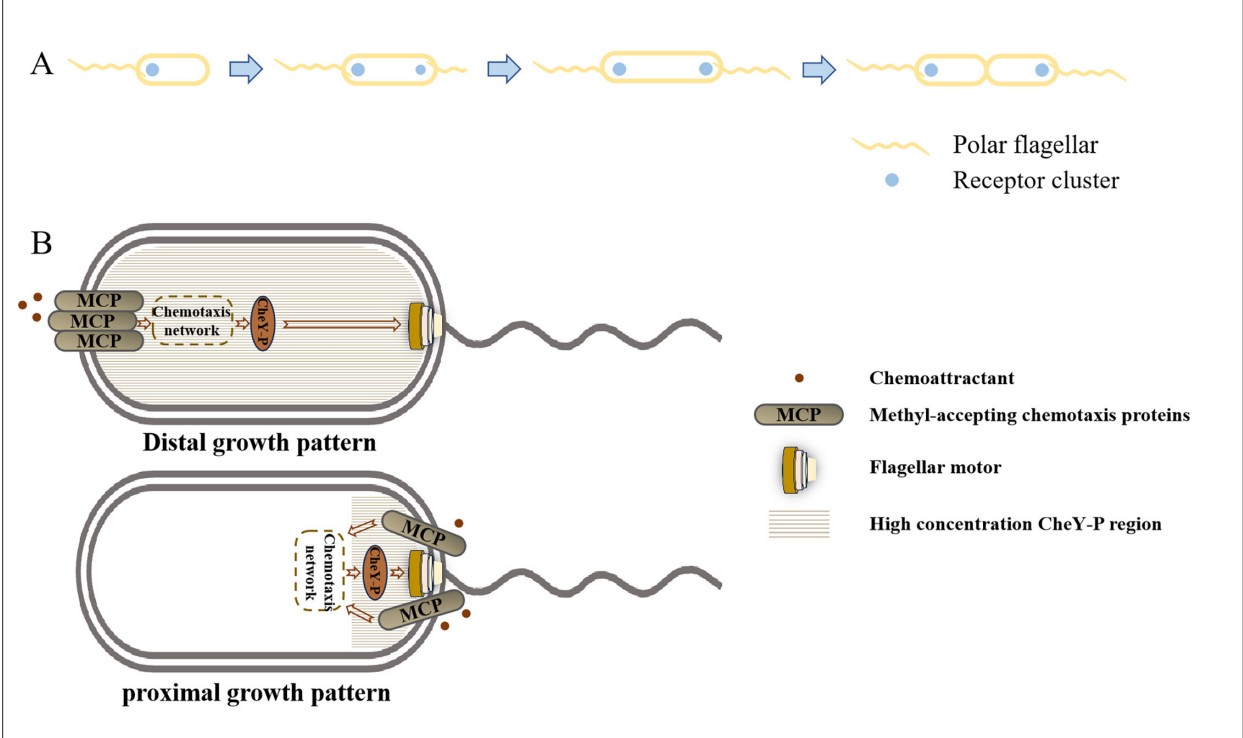

**Figure 5.** The potential physiological significance of this study. (**A**) The construction mode of the chemotaxis network and flagellar motor throughout the complete cell growth cycle. (**B**) The proximal growth mode of the *P. aeruginosa* flagellar motor and receptor clusters will effectively regulate the spatial range of CheY action, thus avoiding unintentional cross-pathway regulation.

membrane curvature factors as seen in *E. coli*. A similar active regulation phenomenon was previously reported in *Vibrio cholerae*, with the corresponding molecular regulation mechanism detailed extensively (*Ringgaard et al., 2014*). The flagella and chemotaxis complex of newborn *V. cholerae* cells are located at the old cell pole, with the polar anchor protein HubP distributed at both cell poles. As the cell grows, HubP recruits the ParA homologue ParC. Subsequently, ParC recruits ParP to assemble a new chemotaxis complex at the new cell pole. Upon cell division, HubP relocates to the middle of the cell, ensuring its robust presences at both poles of the daughter cells. Here, HubP recruits ParA homologues, including FlhG, which influence flagellar motor assembly. However, previous experiments confirmed that the cell cycle-dependent polar localization of ParC and chemotactic proteins is independent of the flagellar position regulatory protein FlhF. This suggests that the localization of flagellar motors and chemotaxis complex in *V. cholerae* is controlled separately through different pathways. In contrast, flagellar motors were no longer robustly distributed at the cell pole after *flhF* was knocked out in *P. aeruginosa*, whereas the assembly site of chemotaxis complex and flagellar motor remained remarkably consistent. This suggests that FlhF regulates the distribution patterns of both and implies an interaction between them.

We further constructed Δ*fliG* and Δ*fliF* mutants, which disrupted the assembly of the flagellar motor. Surprisingly, we observed that the assembly of chemotaxis complex was impaired under these conditions, with a significant decrease in the proportion of individuals exhibiting obvious Che protein fluorescent bright spots. In contrast, the chemotaxis complex assembly in the stator mutant (Δ*motA*Δ*motCD*) was similar to that of the wild-type strain. It should be noted that *P. aeruginosa* has a four-tiered transcriptional regulatory circuit controlling flagellar biogenesis, in which FliG and FliF belong to class II and are regulated by the master transcriptional regulator FleQ (*Dasgupta et al., 2003*). Chemotaxis complex-related genes (*cheA*, *cheW*, etc.) are regulated by intracellular free FliA proteins. In *E. coli*, FliA is bound and inhibited by FlgM, and upon hook assembly, FlgM is secreted outside the cell, allowing FliA to trigger transcription of class III genes (*Chilcott and Hughes, 2000*; *Beeby et al., 2020*), which include the chemosensory genes. This implies that if the hook is not assembled, late genes (including chemoreceptors) should not be expressed. However, Kaplan et al. reported that chemotaxis complexes were observed in *Shewanella oneidensis* Δ*fliF* mutant (*Kaplan et al., 2019*), suggesting a different assembly process from that of *E. coli*. We, therefore, investigated whether knocking out *fliF* or f*liG* affects the expression level of *che* genes in *P. aeruginosa*. Since most experiments in this study focused on CheY, we performed Western blotting to measure CheY expression levels in wild-type, Δ*fliF* and Δ*fliG* strains, and found no significant differences (*Figure 3B*). These results suggest that the observed decrease in the proportion of chemotaxis complexes after knocking out *fliF* and *fliG* should be attributed to the lack of motor assembly rather than changes in Che protein expression levels. To eliminate the potential confounding effects of FlhF and flagellar motors, we constructed a Δ*flhF*Δ*fliF* mutant strain. Under this condition, the proportion of individuals with obvious Che protein fluorescent bright spots dropped to ~50%, indicating that passive regulation based on membrane curvature may still exist in *P. aeruginosa*, albeit to a lesser degree.

Is the remarkable consistency in the spatial distribution of these two independent structural units an unintentional outcome or a hidden mystery? Given that Kulasekara et al. previously proposed that CheA and its phosphorylation state can cause c-di-GMP heterogeneity among individual *P. aeruginosa* cells through a specific PDE (*Kulasekara et al., 2013*; *Kulasekara, 2013*), we sought to verify whether the phenotypic differences induced by CheY here are closely related to changes in the overall population level of c-di-GMP. We introduced a c-di-GMP monitoring system at the single-cell level and discovered that overexpression of CheY substantially increases the intracellular c-di-GMP concentration. Previous studies generally believed that CheY-P, as a chemotactic regulatory protein, only influences flagellar motor switching. Here, we found that CheY-P, as a signaling molecule, can accomplish threshold-limited cross-pathway regulation. At low CheY-P concentrations, it is conservatively involved in motor switching regulation, as confirmed by previous studies. At high CheY-P concentrations, cell motility is suppressed, and c-di-GMP is triggered to express, though its specific mechanism needs further exploration. Unlike CheA, a structural protein of the chemotaxis complex, CheY is a protein that can shuttle in the cytoplasm and can also regulate the intracellular c-di-GMP level. This finding expands our understanding of the connection between the internal chemotaxis network and the c-di-GMP pathway. The intracellular environment is highly crowded (*Zhou et al., 2008*), and the proximal growth pattern shown in *Figure 5B* has multiple physiological significances. On one hand,

chemotaxis-related proteins can precisely regulate cell motility with minimal synthesis costs. On the other hand, the chemotactic network and the flagellar motor are spatially integrated within the cell, ensuring their robust existence within the cell body and preventing mutual interference with collateral pathways.

Many bacteria possess multiple signal transduction pathways. The orderly operation of each pathway within the micrometer-scale space presents a fascinating scientific problem. Here, we utilized the well-known chemotaxis network as a starting point to shed light on this problem, providing inspiration for future in-depth understanding of related issues.

## Materials and methods

### Strains and cell culture

The strains and plasmids used in this study are listed in *Table 1*. The *Escherichia coli* TOP10 strain was used for standard genetic manipulations. A single-colony isolate was grown in 3 ml of LB broth (1% Bacto tryptone, 0.5% yeast extract, and 1% NaCl) overnight to saturation on a rotary shaker (250 rpm) at 37 °C. An aliquot was diluted 1:100 into 10 ml of LB broth and grew to exponential phase. Appropriate antibiotics were added if necessary to prevent plasmid loss: for *E. coli*, 15 µg/ml Gentamicin and 25 µg/ml Tetracycline; for *P. aeruginosa*, 30 µg/ml Gentamicin and 50 µg/ml Tetracycline. To induce protein expression, Isopropyl β-D-1-thiogalactopyranoside (IPTG, 1.0 mM) was added to strains with pME6032-derivative vectors, and arabinose (0.005%, 0.01%, 0.05%, 0.1%) was added to strains with pJN105-derivative vectors. 2 ml of cells were harvested by centrifugation at 2000×$g$ for 2 min, washed twice in an equal volume of motility buffer (MB) [50 mM potassium phosphate, 15 µM EDTA, 0.15 M NaCl, 5 mM Mg$^{2+}$ and 10 mM lactic acid (pH 7.0)] (*Cai et al., 2016*), and resuspended in 5 ml MB for subsequent fluorescence observation.

### Construction of in-frame deletion mutants

We used polymerase chain reaction (PCR) to generate ~1000 bp DNA fragments with upstream (Up) and downstream (Dn) sequences flanking the target gene (including *flhF*, *fliF*, *motA*, and *fliG*) to be deleted. The Up and Dn DNA sequences were fused with the linearized pex18gm vector using Gibson assembly (*Gibson et al., 2009*). The resulting vectors were transformed into *P. aeruginosa* by electroporation, and the desired knockout mutants were obtained by double selections on gentamicin plates (LB plates with 30 µg/ml gentamicin) and sucrose plates (NaCl-free LB plates with 15% sucrose) at 37°C (*Hmelo et al., 2015*).

### Generation of chromosomal fusions of *yfp* to *P. aeruginosa* gene

To generate *eyfp* fusion at the N-terminus of *cheY*, we first used PCR to generate upstream and downstream DNA sequences flanking the *cheY* gene. A subsequent PCR reaction aimed to yield the *cheY-eyfp* fusion with a 3x glycine linker (GGCGGAGGA), with pVS88 serving as the source of the enhanced yellow fluorescent (*eyfp*). The three DNA sequences, along with the linearized vector, were fused to *cheY-eyfp*-pex18gm using Gibson assembly. Finally, gene replacement was used to transfer the fusion constructs into the *P. aeruginosa* chromosome by homologous recombination.

### Flagellar staining and fluorescence imaging

Flagellar filaments were labeled by following the protocol described previously (*Turner et al., 2010*). Cells (1 ml of exponential-phase culture) were harvested by centrifugation at 2000×$g$ for 10 min and washed twice in 1 ml MB. The final pellet was adjusted to a volume of ~100 µl that concentrated the bacterial 10-fold. Alexa Fluor 568 maleimide (Invitrogen-Molecular Probes) was added to a final concentration of 20 µg/ml, and labeling was allowed to proceed for 30 min at room temperature with gyration at 80 rpm. Unused dye was then removed by washing the cells with MB three times, and the final pellet was resuspended in MB.

For fluorescence imaging, 50 µl of cells were added to the chamber (constructed with two double-sticky tapes, a glass slide, and a poly-L-Lysine coated-coverslip), incubated for 3 min, and rinsed with 100 µl MB. The boundary of the chamber was then sealed with Apiezon vacuum grease. The chamber was placed on a Nikon Ti-E inverted fluorescence microscope with a 100× oil-immersion objective and a sCMOS camera (Primer95B, Photometrics). The flagellar filament and chemotaxis complex of cells

**Table 1.** Strains and plasmids used in this study.

| Strain, Plasmid | Genotype, phenotype, and description | Source |
|---|---|---|
| **Strains** | | |
| *P. aeruginosa* | | |
| PAO1 | wild-type strain | Fan Jin Group |
| PAO1 *fliC*<sup>T394C</sup> | Replacement of chromosomal *fliC* in PAO1 | *Tian et al., 2022* |
| PAO1 *fliC*<sup>T394C</sup>*cheY-eyfp* | *yfp* fusions at the N-terminus of *cheY* in PAO1 *fliC*<sup>T394C</sup> | This work |
| PAO1 *fliC*<sup>T394C</sup> *cheY-eyfp* Δ*flhF* | Nonpolar *flhF* deletion in PAO1 *fliC*<sup>T394C</sup> *cheY-eyfp* | This work |
| PAO1 *fliC*<sup>T394C</sup> *cheY-eyfp* Δ*fliF* | Nonpolar *fliF* deletion in PAO1 *fliC*<sup>T394C</sup> *cheY-eyfp* | This work |
| PAO1 *fliC*<sup>T394C</sup> *cheY-eyfp* Δ*fliG* | Nonpolar *fliG* deletion in PAO1 *fliC*<sup>T394C</sup> *cheY-eyfp* | This work |
| PAO1 *fliC*<sup>T394C</sup> *cheY-eyfp* Δ*flhF* Δ*fliF* | Nonpolar *fliF* deletion in PAO1 *fliC*<sup>T394C</sup> *cheY-eyfp* Δ*flhF* | This work |
| PAO1 *fliC*<sup>T394C</sup> *cheY-eyfp* Δ*motA* Δ*motCD* | Nonpolar *motAB* and *motCD* deletion in PAO1 *fliC*<sup>T394C</sup> *cheY-eyfp* | This work |
| PAO1 *fliC*<sup>T394C</sup> Δ*cheA* | Nonpolar *cheA* deletion in PAO1 *fliC*<sup>T394C</sup> | This work |
| PAO1 *fliC*<sup>T394C</sup> *cheY-eyfp* Δ*flgI* | Nonpolar *flgI* deletion in PAO1 *fliC*<sup>T394C</sup> *cheY-eyfp* | This work |
| *E. coli* | | |
| Top10 | F-*mcrA* Δ(*mrr-hsRMS-mcrBC*) Φ80*lacZ*Δ*M15* Δ*lacX74 recA1 ara*D139 Δ(*araleu*)7697 *galU galK rpsL*(Nal<sup>R</sup>) endA1 *nupG* | Invitrogen |
| **Plasmids** | | |
| pex18gm | oriT<sup>+</sup>sacB<sup>+</sup>; gene replacement vector with MCS from pUC18; Gm<sup>r</sup> | Fan Jin Group |
| *flhF*-pex18gm | In-frame deletion of *flhF* cloned into pex18gm; Gm<sup>r</sup> | This work |
| *fliF*-pex18gm | In-frame deletion of *fliF* cloned into pex18gm; Gm<sup>r</sup> | This work |
| *fliG*-pex18gm | In-frame deletion of *fliG* cloned into pex18gm; Gm<sup>r</sup> | This work |
| *motA*-pex18gm | In-frame deletion of *motA* cloned into pex18gm; Gm<sup>r</sup> | This work |
| *motCD*-pex18gm | In-frame deletion of *motCD* cloned into pex18gm; Gm<sup>r</sup> | *Wu et al., 2021* |
| *cheA*-pex18gm | In-frame deletion of *cheA* cloned into pex18gm; Gm<sup>r</sup> | This work |
| *flgI*-pex18gm | In-frame deletion of *flgI* cloned into pex18gm; Gm<sup>r</sup> | This work |
| *cheY-eyfp*-pex18gm | *eyfp* fusions *cheY* cloned into pex18gm; Gm<sup>r</sup> | This work |
| *cheY*-pJN105 | *cheY* overexpression vector in pJN105, *cheY* expression is controlled by P<sub>BAD</sub> promoter; Gm<sup>r</sup> | This work |
| *cheY*-pME6032 | *cheY* overexpression vector in pME6032, *cheY* expression is controlled by P<sub>lac</sub> promoter; Tet<sup>r</sup> | This work |
| *yhjH*-pME6032 | *yhjH* overexpression vector in pME6032, *yhjH* expression is controlled by P<sub>lac</sub> promoter; Tet<sup>r</sup> | This work |
| pCdrA-*gfp* | pUCP22-NotI based cyclic di-GMP level reporter, Gm<sup>r</sup> | *Rybtke et al., 2012* |
| *cheY-eyfp*-pJN105 | *eyfp* fusions *cheY* cloned into pJN105, *cheY-eyfp* expression is controlled by P<sub>BAD</sub> promoter; Gm<sup>r</sup> | This work |
| *cheA-ecfp*-pJN105 | *ecfp* fusions *cheA* cloned into pJN105, *cheA-ecfp* expression is controlled by P<sub>BAD</sub> promoter; Gm<sup>r</sup> | This work |

were observed separately with corresponding filter set, using a 200 ms exposure time. To quantify the fluorescence intensity of the receptor cluster, we took the intensity maximum point as the center and extract a 3×3 pixel matrix around it. The mean value of the elements in this matrix represents the final fluorescence intensity.

## Western blotting

The total protein of *P. aeruginosa* strain cells in each treatment group was extracted and its concentration was measured with Bicinchoninic acid (BCA Protein Quantification Kit, Yeasen Biotechnology Co., Ltd) when the cells grew to exponential phase with same reduced turbidity ($OD_{600}$ between 0.9 and 1.0). The exact amount of protein was subjected to SDS-PAGE electrophoresis, then transferred to a nitrocellulose membrane, which was blocked with 50 mg/mL skim milk in TBST buffer (20 mM Tris, 150 mM NaCl [pH 7.4], 0.1% Tween 20) for 1 hr at room temperature. Since the N-terminal sequence of EYFP is almost identical to that of GFP, a rabbit anti-GFP antibody (1:2000, Yeasen Biotechnology Co., Ltd) and a mouse anti-β-actin antibody (1:2000, Invitrogen) were added and incubated at 4 °C overnight. The nitrocellulose membrane was gently washed with TBST for 20 min, three times. Subsequently, HRP-labeled goat anti-rabbit IgG (1:4000, Abcam) and rabbit anti-mouse IgG (1:4000, Abcam) were added and incubated at room temperature for 1 hr. The nitrocellulose membrane was again gently washed with TBST for 20 min, three times. Then, the membrane was developed using the ECL chemiluminescence detection system (Beyotime P0018FS) and visualized with the ChemiDoc XRS + system (Bio-Rad). Three independent experiments were conducted for reproducibility.

## Monitoring c-di-GMP signal at the single-cell level

For measurements of c-di-GMP signaling, the validated reporter plasmid pCdrA-*gfp*, which produces green fluorescent protein (GFP) in response to an increase in c-di-GMP, was introduced into the experimental system. A 1000×diluted culture was inoculated into the apparatus described in the previous section and allowed to adhere to the surface for ~10 min, with the slight difference that the coverslips were not coated with poly-L-Lysine to avoid interference from background fluorescence. Bacteria were imaged using a Nikon Ti-E inverted microscope, which was equipped with an EMCCD camera (DU897, iXon3, Andor Technology). We used a 100× oil-immersion objective. Bacteria were illuminated with a 488 nm laser (Sapphire 488–200 mW, Coherent), using a standard GFP filter sets in the fluorescence imaging system with an exposure time of 500 ms. To ensure the consistency of culture and observation conditions, the data of wild-type and *cheY*-pME6032 carried strains were collected at the same time. More than 200 independent individuals of each strain were used for data analysis.

Single-cell c-di-GMP concentration was characterized by calculating fluorescence intensity. The average fluorescence intensity of a single cell was calculated by dividing the total fluorescence intensity of a single cell by the cell volume. Bacteria were simplified to a hollow cylindrical trunk with hollow hemispherical caps at both ends. The cell volume can be calculated by taking the length of the short axis of the cell body as the diameter of the sphere and the length of the long axis as the sum of the height of the cylinder and the diameter of the sphere. The mean fluorescence intensity of the cell was obtained by subtracting the average background fluorescence intensity from the total fluorescence intensity of the cell, and then dividing by the cell volume.

## Acknowledgements

This work was supported by National Natural Science Foundation of China Grants (12090053, 12474204, 12304241, and 12304251), a Grant from the Ministry of Science and Technology of China (2019YFA0709303), Grants from the Natural Science Foundation of Shandong Province No. ZR2023QA111 and ZR2023QC168, a special Funding for High-level talents in the Medical and Health Industry in Jinan, a Grant from the Tai Shan Young Scholar Foundation of Shandong Province (tsqnz20231257) and the Science and Technology Development Program of Jinan Municipal Health Commission (2023-1-3).

## Additional information

### Funding

| Funder | Grant reference number | Author |
|---|---|---|
| National Natural Science Foundation of China | 12090053 | Junhua Yuan |

| Funder | Grant reference number | Author |
|---|---|---|
| National Natural Science Foundation of China | 12474204 | Rongjing Zhang |
| National Natural Science Foundation of China | 12304241 | Maojin Tian |
| National Natural Science Foundation of China | 12304251 | Zhengyu Wu |
| Ministry of Science and Technology of the People's Republic of China | 2019YFA0709303 | Rongjing Zhang |
| Natural Science Foundation of Shandong Province | ZR2023QA111 | Zhengyu Wu |
| Natural Science Foundation of Shandong Province | ZR2023QC168 | Maojin Tian |
| Tai Shan Young Scholar Foundation of Shandong Province | tsqnz20231257 | Maojin Tian |
| Science and Technology Development Program of Jinan Municipal Health Commission | 2023-1-3 | Zhengyu Wu |
| Special Funding for High-level talents in the Medical and Health Industry in Jinan | | Zhengyu Wu |

The funders had no role in study design, data collection and interpretation, or the decision to submit the work for publication.

## Author contributions

Zhengyu Wu, Data curation, Software, Formal analysis, Validation, Investigation, Visualization, Methodology, Writing – original draft, Writing – review and editing; Maojin Tian, Sanyuan Fu, Data curation, Software, Formal analysis, Validation, Investigation, Methodology, Writing – original draft; Min Chen, Investigation, Methodology; Rongjing Zhang, Junhua Yuan, Conceptualization, Resources, Supervision, Funding acquisition, Methodology, Writing – original draft, Project administration, Writing – review and editing

## Author ORCIDs

Zhengyu Wu ![ORCID] http://orcid.org/0009-0003-1807-135X
Maojin Tian ![ORCID] https://orcid.org/0009-0004-9302-0162
Rongjing Zhang ![ORCID] http://orcid.org/0009-0008-5519-6385
Junhua Yuan ![ORCID] https://orcid.org/0000-0002-6437-0655

Reviewer #1 (Public review): https://doi.org/10.7554/eLife.97514.4.sa1
Reviewer #2 (Public review): https://doi.org/10.7554/eLife.97514.4.sa2
Reviewer #3 (Public review): https://doi.org/10.7554/eLife.97514.4.sa3
Author response https://doi.org/10.7554/eLife.97514.4.sa4

# Additional files

## Supplementary files
MDAR checklist

## Data availability
All data generated or analysed during this study are included in the manuscript and supporting files.

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
