## [Editor Report · eLife Assessment]

This **important** study by Wu et al presents **convincing** data on bacterial cell organization, demonstrating that the two structures that account for bacterial motility - the chemotaxis complex and the flagella - colocalize to the same pole in *Pseudomonas aeruginosa* cells, and expose the regulation underlying their spatial organization and functioning. This manuscript will be of interest to cell biologists, primarily those studying bacteria.

---

## [Referee Report · Reviewer #1 (Public review)]

Summary:

The study by Wu et al presents interesting data on bacterial cell organization, a field that is progressing now, mainly due to the advances in microscopy. Based mainly on fluorescence microscopy images, the authors aim to demonstrate that the two structures that account for bacterial motility, the chemotaxis complex and the flagella, colocalize to the same pole in *Pseudomonas aeruginosa* cells and to expose the regulation underlying their spatial organization and functioning.

Comments on revisions:

The authors have addressed all major and minor points that I raised in a satisfying way during the revision process. The work can now be regarded as complete: , the assumptions were clarified, the results are convincing, the conclusions are justified, and the novelty has been made clear. This manuscript will be of interest to cell biologists, mainly those studying bacteria, but not only

---

## [Referee Report · Reviewer #2 (Public review)]

Summary:

Here, the authors studied the molecular mechanisms by which the chemoreceptor cluster and flagella motor of *Pseudomonas aeruginosa* (PA) are spatially organized in the cell. They argue that FlhF is involved in localizing the receptors and motor to the cell pole, but a separate mechanism colocalizes them. Finally, the authors argue that the functional reason for this colocalization is to insulate chemotactic signaling from other signaling pathways, such as cyclic-di-GMP signaling.

Strengths:

The experiments and data are high quality. It is clear that the motor and receptors co-localize, and that elevated CheY levels lead to elevated c-di-GMP. The signaling crosstalk argument is plausible.

---

## [Referee Report · Reviewer #3 (Public review)]

Summary:

The authors investigated the assembly and polar localization of the chemosensory cluster in *P. aeruginosa*. They discovered that a certain protein (FlhF) is required for the polar localization of the chemosensory cluster while core motor structures are necessary for the assembly of the cluster. They found that flagella and chemosensory clusters always co-localize in the cell; either at the cell pole in wild type cells or randomly-located in the cell in FlhF mutant cells. They hypothesize that this co-localization is required to keep the level of another protein (CheY-P), which controls motor switching, at low levels as the presence of high-levels of this protein (if the flagella and chemosensory clusters were not co-localized) is associated with high-levels of c-di-GMP and cell aggregations.

Strengths:

The manuscript is clearly-written and straightforward. The authors applied multiple techniques to study the bacterial motility system including fluorescence light microscopy and gene editing. In general, the work enhances our understanding of the subtlety of interaction between the chemosensory cluster and the flagellar motor to regulate cell motility. This work will be of interest to bacteriologists and cell biologists in general.

---

## [Author Response]

The following is the authors’ response to the previous reviews.

**Reviewer #1 (Public review):**
Summary:The study by Wu et al presents interesting data on bacterial cell organization, a field that is progressing now, mainly due to the advances in microscopy. Based mainly on fluorescence microscopy images, the authors aim to demonstrate that the two structures that account for bacterial motility, the chemotaxis complex and the flagella, colocalize to the same pole in *Pseudomonas aeruginosa* cells and to expose the regulation underlying their spatial organization and functioning.Comments on revisions:The authors have addressed all major and minor points that I raised in a satisfying way during the revision process. The work can now be regarded as complete, the assumptions were clarified, the results are convincing, the conclusions are justified, and the novelty has been made clear.This manuscript will be of interest to cell biologists, mainly those studying bacteria, but not only.
**Reviewer #2 (Public review):**
Summary:Here, the authors studied the molecular mechanisms by which the chemoreceptor cluster and flagella motor of *Pseudomonas aeruginosa* (PA) are spatially organized in the cell. They argue that FlhF is involved in localizing the receptors-motor to the cell pole, and even without FlhF, the two are colocalized. Finally, the authors argue that the functional reason for this colocalization is to insulate chemotactic signaling from other signaling pathways, such as cyclic-di-GMP signaling.Strength:The experiments and data are high quality. It is clear that the motor and receptors co-localize, and that elevated CheY levels lead to elevated c-di-GMP.Weakness:The explanation for the functional importance of receptor-motor colocalization is plausible but is still not conclusively demonstrated. Colocalization might reduce CheY levels throughout the cell in order to reduce cross-talk with c-di-GMP. This would mean that if physiologically-relevant levels of CheYp near the pole were present throughout the cell, c-di-GMP levels would be elevated to a point that is problematic for the cell. Clearly demonstrating this seems challenging.

We acknowledge that directly proving the necessity of colocalization to prevent problematic c-di-GMP elevation is experimentally challenging, as it would require creating a system where CheY-P is artificially distributed throughout the cell at physiologically relevant concentrations while maintaining normal chemotaxis function.

However, our data provide several lines of evidence supporting this model. First, we show that CheY overexpression leads to substantial c-di-GMP elevation (71.8% increase) and cell aggregation, demonstrating that elevated CheY levels can indeed cause problematic cross-pathway interference. Second, previous work has shown that CheY-P levels near the pole are an order of magnitude higher than in the rest of the cell (ref. 46). If this elevated CheY-P concentration near the pole were present throughout the cell, our data suggest that c-di-GMP levels would be elevated sufficiently to cause cell aggregation (Fig. 4A), thereby disabling normal motility and chemotaxis. Third, the dose-dependent relationship between CheY concentration and aggregation phenotype supports the idea that precise spatial regulation of CheY levels is functionally important for avoiding cross-pathway interference.

**Reviewer #3 (Public review):**
Summary:The authors investigated the assembly and polar localization of the chemosensory cluster in *P. aeruginosa*. They discovered that a certain protein (FlhF) is required for the polar localization of the chemosensory cluster while a fully-assembled motor is necessary for the assembly of the cluster. They found that flagella and chemosensory clusters always co-localize in the cell; either at the cell pole in wild type cells or randomly-located in the cell in FlhF mutant cells. They hypothesize that this co-localization is required to keep the level of another protein (CheY-P), which controls motor switching, at low levels as the presence of high-levels of this protein (if the flagella and chemosensory clusters were not co-localized) is associated with high-levels of c-di-GMP and cell aggregations.Strengths:The manuscript is clearly written and straightforward. The authors applied multiple techniques to study the bacterial motility system including fluorescence light microscopy and gene editing. In general, the work enhances our understanding of the subtlety of interaction between the chemosensory cluster and the flagellar motor to regulate cell motility.Weaknesses:The major weakness for me in this paper is that the authors never discussed how the flagellar genes expression is controlled in *P. aeruginosa*. For example, in *E. coli* there is a transcriptional hierarchy for the flagellar genes (early, middle, and late genes, see Chilcott and Hughes, 2000). Similarly, Campylobacter and Helicobacter have a different regulatory cascade for their flagellar genes (See Lertsethtakarn, Ottemann, and Hendrixson, 2011). How does the expression of flagellar genes in *P. aeruginosa* compare to other species? how many classes are there for these genes? is there a hierarchy in their expression and how does this affect the results of the FliF and FliG mutants? In other words, if FliF and FliG are in class I (as in *E. coli*) then their absence might affect the expression of other later flagellar genes in subsequent classes (i.e., chemosensory genes). Also, in both FliF and FliG mutants no assembly intermediates of the flagellar motor are present in the cell as FliG is required for the assembly of FliF (see Hiroyuki Terashima et al. 2020, Kaplan et al. 2019, Kaplan et al. 2022). It could be argued that when the motor is not assembled then this will affect the expression of the other genes (e.g., those of the chemosensory cluster) which might play a role in the decreased level of chemosensory clusters the authors find in these mutants.

We thank the reviewer for the valuable suggestions. In the revised manuscript, we have further elaborated on the regulatory control of flagellar genes expression in *P. aeruginosa* (see our response to comment #4).

Comments on revisions:I believe the authors have performed additional experiments that improved their manuscript and they have answered many of my comments and those of the other reviewers. I am supportive of publishing this manuscript, but I still find the following points that are not clear to me (probably I am misunderstanding some points; the authors can clarify).(1) In response to reviewer 1, the authors say that they "analyzed and categorized the distribution of the chemotaxis complex in both wild-type and flhF mutant strains into three patterns: precise-polar, near-polar, and mid-cell localization." I can see what they mean by polar and mid-cell, but near-polar sounds a bit elusive? Can they provide examples of this stage and mention how accurately they can identify it? Also, do the pie charts they show in Figure S4 really show "significant alterations"? There is a difference between 98% and 85% as they mention in their response to reviewer 1, but I am not sure that this is significant? Probably they can explain/change the language in the text? Also, the number of cells they counted for FlhF mutant is more than the double of other strains (WT and FlhF FliF mutant)?

We thank the reviewer for the valuable suggestions. To clarify, we divided the intracellular area along the cell's long axis into three domains: the two ends each representing 10% of the length as the precise-polar domain, the central 50% as the mid-cell domain, and the remaining regions between these as the near-polar domain. The localization pattern of the chemotaxis complex was assigned based on the position of the fluorescence intensity centroid within these domains.

Regarding the significance of the changes, you are correct to question our language. When flhF was knocked out, the proportion of chemotaxis complexes with precise-polar distribution decreased from 98% to 85% - a 13% reduction. While this represents a measurable shift in localization pattern, describing this as "significant alterations" was probably imprecise. We have revised this language to more accurately reflect the magnitude of the change (lines 169-177).

For the cell counting, we increased the sample size for the flhF mutant because this strain exhibited the appearance of mid-cell localization (approximately 5% of cells), which was not observed in wild-type or flhF fliF double mutant strains. To accurately quantify this rare phenotype and ensure statistical reliability, we analyzed more cells for this particular strain. This explains why the flhF mutant dataset contains approximately double the number of cells compared to the other strains.

We have redrawn Figure S4 to include a clear schematic diagram of the cell partitioning method and provided representative examples of each localization pattern (precise-polar, near-polar, and mid-cell) to better illustrate how we distinguished between these categories.

(2) One thing that also confused me is the following: One point that the authors stress is that FlhF localizes both the flagellum and the chemoreceptors to the pole. However, if I look at Figure 2B, the flagellum and the chemoreceptors still co-localize together (although not at the pole). If FlhF was responsible for co-localizing both of them to the pole, then wouldn't one expect them to be randomly localized in this mutant and by that I mean that they do not co-localize but that each of them (the flagellum and the chemoreceptors) are located in a different random location of the cell (not co-localized). The fact that they are still co-localized together in this mutant could also be interpreted by, for example, that FlhF localizes the flagellum to the pole and another mechanism localizes the chemoreceptors to the flagellum, hence, they still co-localize in this mutant because the chemoreceptors follow the flagellum by another mechanism to wherever it goes?

Thank you for this insightful observation. You are correct that our current experimental results do not definitively establish that FlhF directly localizes both the flagellum and chemoreceptors to the pole independently. The persistent colocalization of flagella and chemoreceptors in the DflhF mutant, even when both are mislocalized away from the pole, actually suggests a more complex regulatory mechanism than we initially proposed.

This observation highlights an important distinction between polar targeting and colocalization maintenance. Our data suggest that FlhF influences the polar targeting of the flagellum-chemoreceptor assembly, but the colocalization itself appears to be governed by a different mechanism that operates independently of FlhF. This could involve direct protein-protein interactions between flagellar and chemotaxis components, or shared assembly machinery that we have yet to identify.

To better reflect this interpretation, we have revised the subsection title (line 150). We have also modified the relevant discussion (line 180) to more accurately describe FlhF’s role in polar targeting rather than claiming it directly controls chemoreceptor localization.

(3) In the response to reviewers, the authors mention "suggesting that the assembly of the receptor complex is likely influenced mainly by the C-ring and MS-ring structures rather than by the P ring". However, in the article, they still write "The complete assembly of the motor serves as a partial prerequisite for the assembly of the chemotaxis complex, and its assembly site is also regulated by the polar anchor protein FlhF" despite their FlgI results which is not in accordance with this statement? Also, As I mentioned in my previous report, in FliG and FliF mutant the motor does not assemble (see Hiroyuki Terashima et al. 2020., and Kaplan et al., 2022).

We thank the reviewer for the suggestions and acknowledge the contradictions in our original text. You are correct that in DfliF and DfliG mutants, the flagellar motor does not assemble, while the P ring (FlgI) functions as a bushing for the peptidoglycan layer and its absence does not prevent motor assembly.

Our DflgI results, which showed normal chemotaxis complex assembly similar to wild-type, clearly demonstrate that the P ring is not required for chemoreceptor complex formation. This contradicts our original statement that "complete assembly of the motor serves as a partial prerequisite for the assembly of the chemotaxis complex."

We have corrected this inconsistency by: (1) Revising the subsection title (line 186) to more accurately reflect that core motor structures, rather than complete motor assembly, influences chemoreceptor complex formation. (2) Modifying sentences in the introduction (lines 97-98) to better align with our experimental findings.

(4) The authors have said in their response to my point "and currently, there is no evidence that FliA activity is influenced by proteins like FliG". I just want to clarify what I meant in my previous report: In *E. coli*, FliA binds to FlgM, and when the hook is assembled FlgM is secreted outside the cell allowing FliA to trigger the transcription of class III genes, which include the chemosensory genes (see Figure 5 in Beeby et al, 2020 in FEMS Microbiology, and Chilcott and Hughes, 2000). This implies that if the hook is not built, then late genes (including the chemoreceptors) should not be present. However, in Kaplan et al., 2019, the authors imaged a FliF mutant in Shewanella oneidensis (Figure S3) and still saw that chemoreceptors are present (I believe the authors must highlight this). This suggests that species such as Shewanella and Pseudomonas have a different assembly process than that *E. coli*, and although the authors say that in the text, I believe they still can refine this part more in the spirit of what I wrote here.

We thank the reviewer for the important clarification regarding the differences in transcriptional regulation among bacterial species. We agree that the observation of chemoreceptors in Shewanella oneidensis DfliF mutants (Kaplan et al., 2019) represents a significant deviation from the well-characterized *E. coli* model and merits stronger emphasis. In response, we have expanded the discussion to more clearly highlight the critical distinctions in the transcriptional regulatory circuits governing flagellar and chemoreceptor biogenesis between *E. coli* and species such as Shewanella oneidensis and *Pseudomonas aeruginosa* (lines 351-363).

I do not like to ask for additional experiments in the second round of review, so for me if the authors modify the text to tackle these points and allow for probable alternative explanations/ highlight gaps/ modify language used for some claims, then that is fine with me.
**Reviewer #2 (Recommendations for the authors):**
It is plausible that colocalization reduces CheY levels throughout the cell in order to reduce cross-talk with c-di-GMP. This would mean that if physiologically-relevant levels of CheYp near the pole were present throughout the cell, c-di-GMP levels would be elevated to a point that is problematic for the cell. Clearly demonstrating this seems challenging.

We acknowledge that directly proving the necessity of colocalization to prevent problematic c-di-GMP elevation is experimentally challenging, as it would require creating a system where CheY-P is artificially distributed throughout the cell at physiologically relevant concentrations while maintaining normal chemotaxis function.

However, our data provide several lines of evidence supporting this model. First, we show that CheY overexpression leads to substantial c-di-GMP elevation (71.8% increase) and cell aggregation, demonstrating that elevated CheY levels can indeed cause problematic cross-pathway interference. Second, previous work has shown that CheY-P levels near the pole are an order of magnitude higher than in the rest of the cell (ref. 46). If this elevated CheY-P concentration near the pole were present throughout the cell, our data suggest that c-di-GMP levels would be elevated sufficiently to cause cell aggregation (Fig. 4A), thereby disabling normal motility and chemotaxis. Third, the dose-dependent relationship between CheY concentration and aggregation phenotype supports the idea that precise spatial regulation of CheY levels is functionally important for avoiding cross-pathway interference.